# Kikwit Ebola Virus Disease Progression in the Rhesus Monkey Animal Model

**DOI:** 10.3390/v12070753

**Published:** 2020-07-14

**Authors:** Richard S. Bennett, James Logue, David X. Liu, Rebecca J. Reeder, Krisztina B. Janosko, Donna L. Perry, Timothy K. Cooper, Russell Byrum, Danny Ragland, Marisa St. Claire, Ricky Adams, Tracey L. Burdette, Tyler M. Brady, Kyra Hadley, M. Colin Waters, Rebecca Shim, William Dowling, Jing Qin, Ian Crozier, Peter B. Jahrling, Lisa E. Hensley

**Affiliations:** 1Integrated Research Facility, Division of Clinical Research, National Institute of Allergy and Infectious Diseases, National Institutes of Health, 8200 Research Plaza, Frederick, MD 27102, USA; James.Logue@som.umaryland.edu (J.L.); xianhong.liu@nih.gov (D.X.L.); rebecca.reeder@nih.gov (R.J.R.); krisztina.janosko@nih.gov (K.B.J.); donna.perry@nih.gov (D.L.P.); timothy.cooper@nih.gov (T.K.C.); byrumr@niaid.nih.gov (R.B.); raglandd@niaid.nih.gov (D.R.); stclairem@niaid.nih.gov (M.S.C.); ricky.adams@nih.gov (R.A.); tracey.burdette@nih.gov (T.L.B.); tyler.brady@nih.gov (T.M.B.); kyra.hadley@nih.gov (K.H.); mason.waters@nih.gov (M.C.W.); rebecca.shim@nih.gov (R.S.); jahrlingp@niaid.nih.gov (P.B.J.); lisa.hensley@nih.gov (L.E.H.); 2Research Resources Section, Office of Biodefense, Research Resources, and Translational Research/Division of Microbiology and Infectious Diseases, National Institute of Allergy and Infectious Diseases, National Institutes of Health, 5601 Fishers Lane, Rockville, MD 20892-9825, USA; william.dowling@cepi.net; 3Biostatistics Research Branch, National Institute of Allergy and Infectious Diseases, National Institutes of Health, 5601 Fishers Lane, Rockville, MD 20892, USA; jingqin@niaid.nih.gov; 4Clinical Monitoring Research Program Directorate, Frederick National Laboratory for Cancer Research, Frederick, MD 21702 USA; ian.crozier@nih.gov; 5Emerging Viral Pathogens Section, National Institute of Allergy and Infectious Diseases, National Institutes of Health, 8200 Research Plaza, Frederick, MD 21702, USA

**Keywords:** Ebola virus, Kikwit, FANG, rhesus monkey, natural history, emerging pathogens, risk group 4, BSL-4, animal model

## Abstract

Ongoing Ebola virus disease outbreaks in the Democratic Republic of the Congo follow the largest recorded outbreak in Western Africa (2013–2016). To combat outbreaks, testing of medical countermeasures (therapeutics or vaccines) requires a well-defined, reproducible, animal model. Here we present Ebola virus disease kinetics in 24 Chinese-origin rhesus monkeys exposed intramuscularly to a highly characterized, commercially available Kikwit Ebola virus Filovirus Animal Non-Clinical Group (FANG) stock. Until reaching predetermined clinical disease endpoint criteria, six animals underwent anesthesia for repeated clinical sampling and were compared to six that did not. Groups of three animals were euthanized and necropsied on days 3, 4, 5, and 6 post-exposure, respectively. In addition, three uninfected animals served as controls. Here, we present detailed characterization of clinical and laboratory disease kinetics and complete blood counts, serum chemistries, Ebola virus titers, and disease kinetics for future medical countermeasure (MCM) study design and control data. We measured no statistical difference in hematology, chemistry values, or time to clinical endpoint in animals that were anesthetized for clinical sampling during the acute disease compared to those that were not.

## 1. Introduction

Ebola virus (EBOV) has caused sporadic outbreaks in Middle and Western Africa with an average lethality rate ranging of 44% [1]. Until the Ebola virus disease (EVD) outbreak in Western Africa (2013–2016) and more recently the 2018–2020 EVD outbreak in the Democratic Republic of the Congo, testing therapeutics during infection had relied heavily on the use of animal models. Although rodent animal models of EVD are available, they require adapting EBOV isolates to replicate and cause disease in either laboratory mice or domesticated guinea pigs and some aspects of human disease cannot be recapitulated [2,3,4,5,6,7,8,9]. More recently, laboratory mice seeded with human immune, liver, and thymus cells supported virus replication leading to lethal disease. However, some variation in disease severity was noted between mice with different human cell donors [10,11]. Domestic ferrets are permissive to infection and exhibit many signs of human EVD including rash and gastrointestinal signs [12,13,14]. Laboratory models of EVD are often used for medical countermeasures (MCMs) testing, which often tests MCM in two animals models when available. As a result, lethal nonhuman primates’ infection models that recapitulate the most severe aspects of human disease have long been a “gold standard” in MCM testing. Of the two most commonly used nonhuman primates in EBOV research, rhesus monkeys and cynomolgus macaques, rhesus monkeys have a longer terminal disease period (5–9 days, average 6.5) compared to cynomolgus macaques (6–7 days) [15,16,17,18]. The prolonged disease course in rhesus monkeys provides a longer window for testing medical countermeasures as well as understanding EBOV pathogenesis.

In an effort to standardize the virus used to test medical countermeasure efficacy, a commercially available and standardized Kikwit variant of EBOV (EBOV/Kik) allows maximum containment laboratories (Biosafety Level 4 laboratory in the U.S.A.) to use the same virus in animal studies, thus, allowing easy data comparisons across laboratories. The goal of this study was three-fold. First, the study was designed to measure if sample collection under anesthesia impacted how quickly animals reached predetermined clinical endpoint criteria. Second, this study was to measure clinical pathology kinetics following rhesus monkey exposure to a highly characterized, commercially available challenge virus. Finally, this study generated an expansive library of tissue samples for use by current and future study collaborators. The data and samples generated in this study are intended to help guide future medical countermeasure study design.

## 2. Materials and Methods 

### 2.1. Virus 

Ebola virus/H.Sapiens-tc/COD/1995/Kikwit-9510621 (EBOV/Kik, GenBank accession number MG572235.1) was obtained through Biodefense and Emerging Infections Research Resources Repository (BEI Resources) (catalog number NR-50306). No additional passaging was performed at the Integrated Research Facility at Fort Detrick. All work with infectious EBOV/Kik was performed in the Biosafety Level 4 laboratory at the Integrated Research Facility at Fort Detrick, Maryland, USA.

### 2.2. Animals 

Twenty-seven (11 male and 16 female), 5- to 6-year-old Chinese-origin rhesus monkeys were randomized balancing age, weight, and sex across seven groups (Appendix A). Three animals were not infected and served as uninfected control sample and tissues sources and 24 animals were infected with virus. Two baseline blood samples were collected from all study animals at approximately 30 (baseline 1, B1) and 14 (baseline 2, B2) days prior to EBOV exposure. Animals assigned to infection groups were exposed to a target dose of 1000 PFU EBOV/Kikwit in the left lateral triceps muscle diluted in a total volume of 1 mL at the study day 0 (exposure day). All animals were observed and clinically scored for signs of EVD up to three times daily post-EBOV exposure. Individual body weights and food consumption were recorded daily. Blood samples were collected from all animals at scheduled time points (Figure 1). Uninfected controls (*n* = 3) were humanely euthanized and necropsied at baseline while EBOV-exposed monkeys were serially euthanized and necropsied at three days post-exposure (DPE, *n* = 3), four DPE (*n* = 3), five DPE (*n* = 3), six DPE (*n* = 3), or when they reached predetermined clinical endpoint criteria (*n* = 12) with EVD. Animals allowed to reach clinically predetermined endpoints with EVD were further split into two groups: Animals from which serial swabs, blood, and semen samples were collected under anesthesia during the disease course (*n* = 6) and animals that had no anesthesia or in-life samples collected post-EBOV-exposure (*n* = 6). Challenge groups 1–4 were exposed to virus on separate days due to logistic reasons. Each exposure day used a fresh vial of virus. The study design is shown in Figure 1.

### 2.3. Animal Ethics’ Disclosure 

This study was carried out in strict adherence to the Guide for the Care and Use of Laboratory Animals of the National Institute of Health, the Office of Animal Welfare, and the US Department of Agriculture. All work was approved by the National Institute of Allergy and Infectious Diseases (NIAID) Division of Clinical Research Animal Care and Use Committee and performed at the NIAID Research Facilities. Clinical procedures were carried out after animals had been anesthetized by trained personnel under the supervision of veterinary staff. Food and water were available ad libitum.

### 2.4. Reverse-Transcriptase Quantitative Real-Time PCR (RT-qPCR)

Total RNA was isolated as described previously [19]. Briefly, 70 µL of sample inactivated by Trizol LS was added to 280 µL of viral lysis buffer (AVL) (Qiagen, Germantown, MD, USA) with carrier RNA. Samples were then extracted using the QIAamp Viral RNA Mini Kit (Qiagen) in accordance with the manufacturer’s instructions, eluted in 70 µL of elution buffer (AVE) (Qiagen), aliquoted, and frozen. Viral RNA titer was determined using an experimental BEI Resources Critical Reagents Program EZ1 qRT-PCR kit assay in accordance with the manufacturer’s instructions. Sample titers were reported as RNA genome equivalents (GE) per mL of sample.

### 2.5. GeneXpert 

Samples were tested using the Cepheid GeneXpert. Plasma samples were processed according to the manufacturer’s instructions and semen samples were processed as described previously [20]. Samples with a failed internal control (SAC, sample adequacy control), but passing internal controls (CIC, Cepheid internal control), were considered negative due to limited cross-reactivity with human and nonhuman primate housekeeping genes.

### 2.6. Plaque Assay 

Each test sample was serially diluted 1:10 for a total of 3 dilutions in alpha-Minimum Essential Medium (α-MEM) enriched with 10% Heat inactivated-fetal bovine serum (HI-FBS) and antibiotic-antimycotic solution (Gibco, ThermoFisher Scientific) to a final concentration of 100 units/mL of penicillin, 100 µg/mL of streptomycin, and 0.25 µg/mL of amphotericin B. Three-hundred µL from each sample dilution was then added to a 90% confluent monolayer of Vero E6 cells (BEI #NR-596) and incubated at 37 °C and 5% CO_2_ for 1 h with rocking every 10 ± 5 min. After incubating, a 1:1 ratio of 2.5% Avicel biopolymer (FMC Corporation) and Temin’s modified Eagle medium (ThermoFisher Scientific) supplemented with 10% FBS, antibiotic-antimycotic solution, and L-alanyl-L-glutamine was overlaid, and plates were incubated at 37 °C and 5% CO2 for 8 days. After the 8-day incubation, the Avicel overlay was removed and a 0.2% crystal violet stain (Ricca Chemical, 3233-16, Pocomoke City, MD, USA) prepared in 10% neutral buffered formalin (Thermo Scientific, Waltham, MA, USA 5705) was added to each well and incubated at room temperature for 30 min. Cell monolayers were then washed with water, and plaques were manually counted.

### 2.7. Tissue Processing 

Tissue samples collected at necropsy were frozen in bead beating tubes for homogenization and inactivation. Samples were thawed and processed by adding TRIzol (Ambion, Thermo Fisher, Waltham, MA, USA) to create a final 10% *w*/*w* homogenate and homogenized 2 consecutive times for 90 s at 500 rpm using the Omni BeadRuptor instrument (Omni International, Tulsa, OK, USA). The tissue homogenates were then inactivated in TRIzol LS and removed from the Biosafety level 4 (BSL-4) laboratory.

### 2.8. Serum Processing 

Blood samples were collected in BD vacutainer serum separator tubes (Becton Dickinson, Franklin Lakes, New Jersey, USA) and incubated at room temperature for at least 30 min prior to being centrifuged at room temperature for 10 min at 1800× *g*. The separated serum was then transferred into a clean 2-mL Sarstedt tube. The sample was evaluated for the presence of hemolysis, icterus, and/or lipemia and the presence or absence of these sample conditions was recorded. One hundred µL of the serum sample were then transferred into a Piccolo General Chemistry 13-panel cartridge (Abaxis, Inc., Union City, CA, USA) and a Piccolo General Renal Function Panel cartridge (Abaxis, Inc., Union City, CA, USA) within 20 min of the removal of the cartridge from a 4 °C refrigerator. The cartridge was then immediately loaded into the Abaxis Piccolo analyzer. All internal quality controls were verified manually before sample results were reported.

### 2.9. Hematology 

Blood samples were collected in BD vacutainer plastic blood collection tubes with tripotassium ethylenediaminetetraacetic acid (K3EDTA) (Becton Dickinson, Franklin Lakes, NJ, USA) and mixed by gentle inversion. Each sample was examined for clots with wooden applicator sticks and any excessive clotting was noted. Then, 250 µL of each sample were then transferred to individual clean 0.5-mL Sarstedt tubes and analyzed on a Sysmex XT-2000iV (Sysmex Corporation, Kobe, Hyogo Prefecture, Japan) following an initial quality check on the instrument.

### 2.10. Semen Collection

Semen/ejaculate samples’ collection was attempted from non-vasectomized, anesthetized male nonhuman primates (NHPs) using an Electro Ejaculator (Minitube). The NHP was anesthetized with ketamine (10 to 15 mg/kg) by intramuscular (IM) injection, and a 6.35 cm lubricated rectal probe was inserted into the rectum until the tip of the probe was palpated abdominally just cranial to the pubic symphysis. The power source was connected to the probe and placed on automatic mode, which increases the power output in increments of 0.5 V from 0 to 10 V, and in steps of 1.0 from 10 to 20 V, reaching a maximum of 20 V, in pulses at 2-s intervals. Pulses and intervals were adjusted as needed from automatic, if a semen sample was not obtained, to manual, which generates continual or intermittent voltage adjustment and pulsing. During ejaculation, each sample was collected in an individual 50-mL conical tube, which was directly submitted for analysis. If ejaculation was not achieved after two complete cycles, collection efforts ceased.

### 2.11. Clinical Observations, Euthanasia Criteria, and Necropsy 

NHPs were observed cage-side by study personnel and documented at least once per day. Animals were assigned a clinical score based on five criteria: Overall clinical appearance; respiratory rate, mucous membrane color, dyspnea; recumbency; non-responsiveness; and core body temperature (Appendix A). Animals were observed once daily until at least one animal progressed to a clinical score of 3, at which time a second, officially documented observation was performed in the afternoon. When at least one study animal achieved a clinical score of 5, a third observation was added and officially documented. If any animal reached a clinical score equal to or higher than 10, the animal was humanely euthanized. Complete necropsies were conducted on all animals. Gross pathological findings were recorded and the tissue samples collected were fixed in 10% neutral buffered formalin (NBF) for further processing.

### 2.12. Statistical Methods 

The two-sample t-test was used to compare the minimum or maximum values collected. A *p*-value threshold of 0.01 was used to assess statistical significance. All statistical analyses were performed in S-plus (TIBCO Software Inc., Palo Alto, CA, USA).

## 3. Results

### 3.1. Virus Inoculum, Viremia, and Virus Detection in the Oral Cavity

Viral inoculum for each exposure group was titered by Avicel plaque assay at the time of animal exposure. Inoculum titers were 8.67 × 10^2^, 1.80 × 10^3^, 1.09 × 10^3^, and 1.29 × 10^3^ PFU for cohorts 1–4, respectively.

Following virus exposure, all animals became viremic at 3 DPE, and remained viremic through 8 DPE as determined by RT-qPCR (Figure 2a), although genomic material could be detected by Cepheid GeneXpert as soon as 2 DPE from whole blood samples (Figure 2c). Following virus exposure, only two animals, NHP20 and NHP24, were considered negative on day 2 due to an invalid internal control but passing quality controls. Plaque assay was the least sensitive method for virus detection with all animals’ plaque assays positive by day 5 DPE (Figure 2b). Viral RNA could only be detected from oral swabs during late disease (from day 5 onward) (Appendix A).

### 3.2. Clinical Observations, Scoring, and Survival

Animals began showing clinical signs of EVD by 4 DPE as determined by cage-side clinical scoring and continued to show signs of disease through 8 DPE (Figure 3a). Food consumption started to decline at DPE 1 and was significantly low at the end-stage of EBOV disease (Figure 3b). Average biscuit consumption was 65.10%, 68.53%, 73.65%, 43.69%, 5.27%, 1.56%, 0.97%, and 1.61% from DPE 1 to 8, respectively, when compared to biscuit consumption on DPE 0 (Figure 3b). Skin rashes (petechia) were observed on the face and arms from DPE 5–8 (Appendix A).

All animals that were allowed to reach predetermined clinical endpoint criteria were euthanized from DPE 5–8 (Figure 3c). The average survival time post-EBOV-exposure was 164 h (no sample collection following exposure) and 172 h (with sample collection following exposure). There were no statistically significant differences in body temperature (while under anesthesia), clinical score, food consumption, body weight, or survival time (*p* = 0.60 by *t*-test and *p* = 0.56 by Wilcoxon) between animals anesthetized for blood collection during the disease course (*n* = 6) and animals that had no anesthesia or in-life clinical sample collections performed post-EBOV-exposure (*n* = 6). In addition, there was no difference in survival time between male or female animals anesthetized for blood collection during the disease course and male and female animals that had no anesthesia (males *p* = 0.1628, females *p* = 0.1912 by *t*-test).

### 3.3. Serum Chemistry

Serum chemistry results were generally within normal reference interval (RI) values for nearly all analytes evaluated until DPE 4 (Table 1, individual values in Appendix A). The mean and individual activity of alanine aminotransferase (ALT) increased sharply above the RI (20–58 U/L) and were 5.32-, 8.31-, 14.30-, and 10.42-fold over the mean baselines from DPE 5–8, respectively. On DPE 3, four of 12 macaques sampled had aspartate aminotransferase (AST) activities at or slightly above RI (22–45 U/L), with a single animal double the normal upper limit (97 U/L). By DPE 5, nine of 10 macaques sampled had AST activities 3.9–31 times the upper reference limit, with continuing increases until termination. Serum concentrations of gamma–glutamyl transpeptidase (GGT) were within RI (25–84 U/L for females and 40–125 for males) up to DPE 5, except for the single animal that reached clinical endpoint criteria on that day. GGT was elevated in all but two females on DPE 6 and elevated in all animals by DPE 7–8. Alkaline phosphatase (ALP) concentrations paralleled GGT. Total bilirubin (TBIL) concentrations began to increase at DPE 4 and were elevated above RI (0.3–0.6 mg/dL) in all but two animals by termination at DPE 6–8. Increased total bilirubin, GGT, and alkaline phosphatase were all consistent with cholestasis in terminal disease as liver function became significantly impaired (associated with viral-associated hepatocellular degeneration and necrosis).

Serum concentrations of albumin (ALB) were below RI (3.0–4.0 g/dL) in nine of 10 animals on DPE 5. All animals were hypoalbuminemic at termination of DPE 6–8, with profound hypoalbuminemia (<2.0 g/dL) in three animals. Changes in serum total protein (TP) and globulins paralleled those of albumin.

By DPE 5, blood urea nitrogen (BUN) in two macaques rose sharply (1.70- and 5.00-fold over baseline measures), although the average BUN was within the RI (12–25 mg/dL). Mean BUN increased over the RI during EVD progression and was 1.55-, 6.19-, and 3.29-fold over the mean baseline (BL0) at DPE 6–8, respectively. Serum concentrations of creatinine (CRE) began to increase for most animals at DPE 5, and all animals except one had elevated creatinine in terminal samples on DPE 6–8 (consistent with prerenal azotemia (dehydration)). Three animals were profoundly azotemic (BUN > 140, CRE > 8) at termination on DPE 7 and 8.

Serum calcium (CAL) concentrations began to decrease as early as DPE 3 and continued to decline inexorably for all but two animals as the study progressed. Eight of 10 animals were hypocalcemic on DPE 5, and all but one was hypocalcemic on DPE 6–8. Three animals were significantly hypocalcemic (<6.0 g/dL) at termination on DPE 7. Correction of serum calcium for hypoproteinemia did not alter these trends. Serum phosphorus concentrations were generally unremarkable, with increases in four animals at termination. One of these animals also had a calcium x phosphorus product in excess of 70.

The anion gap increased and total CO_2_ decreased in most (but not all) terminal samples on DPE 6–8, consistent with metabolic (lactic) acidosis due to hypoperfusion/hypovolemia. Serum chloride concentrations consistently decreased during the course of EVD, with terminal values below RI (102–11) for 10 of 12 macaques on DPE 6–8. Sodium concentrations often, but not invariably, paralleled chloride. The sodium–chloride difference was generally within RI for all animals throughout the study. Loss of sodium and chloride is consistent with hypotonic dehydration. Serum concentrations of potassium (K^+^) generally decreased from DPE 1–5, consistent with inanition. However, K^+^ concentrations were markedly variable among macaques terminally, including three macaques that were hyperkalemic at termination on DPE 7 (samples not hemolyzed). Hyperkalemia in these animals (and rebounding K^+^ concentrations in additional animals) correlated with metabolic acidosis, indicating an intracellular to extracellular shift of K^+^ ions in exchange for H^+^ ions.

Amylase (AMY) concentrations were markedly variable at the end stage of disease between macaques and time points, with several animals below RI (inanition) and a few animals above. There were no statistically significant differences between terminal animals with or without routine manipulations during disease course for these parameters. Average glucose (GLU) levels remained within the reference range until DPE 8.

### 3.4. Hematology

Hematology data were collected twice prior to virus exposure and periodically following exposure (Table 2, individual values in Appendix A). On average, animals developed a monocytosis on DPE 3 and monocytopenia on DPE 7, returning to within the reference range between those study days. Animals developed a neutrophilia from DPE 3–5 and platelets dropped to below reference ranges from DPE 5–8. No other hematological parameters generally varied outside the Integrated Research Facility (IRF) -Frederick-defined normal RIs for Chinese-origin rhesus monkeys. There were no statistically significant differences between terminal animals with or without routine manipulations during disease course for these parameters.

Leukocyte counts were generally unremarkable until DPE 3. Total white blood cell and absolute neutrophil counts increased transiently from DPE 3 to 4 or 5 (peak), and then dropped sharply (but were within the RI) from DPE 5 to 8. Only a single animal was leukopenic and neutropenic, while two animals had neutrophilia (one with leukocytosis), at termination on DPE 6–8. Similarly, absolute monocyte counts increased transiently on DPE 3 then dropped sharply, generally to below the RI (100–690/µL). Ten of 18 terminal samples from DPE 5 or later were monocytopenic.

Generally, absolute lymphocyte numbers began to decline on DPE 3, reaching a nadir as the disease progressed. Four macaques were lymphocytopenic on DPE 4 or 5. In four animals, lymphocyte numbers were increasing in the terminal samples. Consistent with these findings, the neutrophil:lymphocyte ratio increased from DPE 3–6 for most animals before returning toward normal.

Platelet counts began a downward trend as early as DPE 2 and continued to decrease with EVD progression. Platelet counts were below RI (202–465 × 10^3^/µL) for all animals from DPE 5 onward, and 11 macaques were thrombocytopenic (<100 × 10^3^/µL) at termination.

Generally, macaques showed mild to moderate decrease in red blood cell count (RBC#) in both males and females from DPE 3–8, with terminal rebounds (interpreted as blood loss (hemorrhage) followed by hemoconcentration (dehydration)). Similar trends were present with hematocrit and hemoglobin.

Reticulocyte count (retic#) decreased as early as DPE 2 in females and three males and continued to drop severely until the end of the study.

The progressive decrease in reticulocytes was consistent with an early shift toward myelopoiesis and terminal exhaustion of the bone marrow reserves.

### 3.5. EBOV Detection in Semen

Of six animals that reached endpoint criteria, semen/ejaculate collection was successful in five, with cycle threshold (CT) values ranging from 32.1–37.3 for the Ebola virus glycoprotein (GP, GeneExpert) and 29.1–35.5 for the Ebola virus nucleoprotein (NP, GeneExpert) (Table 3). CT values for EBOV NP gene targets were consistently lower than CT values for EBOV GP in all animals.

## 4. Discussion

Here we describe the disease progression of EVD in rhesus monkeys using a highly characterized EBOV challenge stock. As comparisons between studies can often be confounded by the use of varying experimental conditions (e.g., virus stock, challenge route, blood collection practices), the goal of the present study was to provide a detailed description of EVD progression using a commercially available and well-characterized EBOV challenge stock. This description will allow laboratories across the globe to compare to this control data set while utilizing the same virus preparation during MCM testing. Importantly, a number of collaborators on this project will be publishing data on varying aspects of immunology and disease process using additional samples collected from the present study, further adding to the complete description of disease progression. The combination of these data, and the use of the commercially available EBOV stock, will hopefully provide a model for future data comparison across labs.

EVD progression in animals on this study was similar to previously reported studies [21,22,23]. The first indication of infection was detection of the viral NP gene in blood at two days post-exposure (summary in Figure 4). By DPE 3, all animals were viremic with a steady increase in titer between DPE 3–5, after which the titers appeared to plateau. All animals remained viremic until they met end-point criteria for euthanasia. Viremia is the first marker of infection and was detected prior to changes in hematology parameters, serum chemistry analytes, or clinical signs. 

Starting on DPE 3, increases in monocytes, neutrophils, and overall white blood cells (WBCs) were measured. Increases in these WBCs were sustained until DPE 5, after which they returned to within reference intervals. 

By DPE 4, food intake decreased and changes in blood hematology were present prior to the animals exhibiting cage-side signs of infection. Clinical scores continued to increase until animals reached clinical endpoint criteria. In this study, predefined clinical endpoint criteria were used to determine euthanasia time points in a subset of the groups (terminal necropsy groups with and without routine sample collection), allowing morbidity to serve as a surrogate for lethality. The first animal met end-point criteria on the morning of DPE 5, while other animals reached clinical end-point criteria by DPE 8. During the time when animals reached terminal disease, a rapid decline in platelets and reticulocytes was measured as well as changes in blood chemistries (ALT, ALB, ALP, AST, BUN, CAL, CRE, GGT, GLU, TBIL, and TP).

Under BSL-4 conditions, animals must be anesthetized prior to collection of these clinical hematology and chemistry samples. To determine if anesthesia for clinical sampling during acute EVD impacted survival time, two groups of animals were allowed to reach clinical end-point criteria. Of these two groups, one group was anesthetized for serial clinical sample collections, whereas the other group underwent no in-life sample collection during the disease course. Under these conditions, the routine clinical sampling under anesthesia did not negatively impact the survival time measured by time-to-end-point criteria, hematology, or biochemical analysis.

Here we describe Ebola virus infection and disease in a nonhuman primate model using a highly characterized viral stock. Unlike humans, the disease time-course in nonhuman primates is highly truncated and almost universally lethal. Following exposure, the earliest indicators of disease could only be identified through PCR for EBOV in serum and hematology changes, but not by cage-side observations. These data indicate that early markers of disease, measured via blood collection, did not influence survival time. However, the study design allowed for a 48-h recovery period between sedation periods and only one blood collection during terminal disease. Daily, or more frequent blood collection, would likely only be possible through the use of implanted catheters due to concerns that frequent sedation could impact animal behavior or feeding and drinking during critical disease time points. In addition, as organ function begins to decrease during infection, eliminating the sedatives from the body becomes more difficult for the animals. This study identified that routine, short-duration anesthesia required for sample collections did not negatively impact Ebola virus disease course. Ongoing research is focused on how the earliest marker of disease, viremia, quickly establishes a systemic infection in a wide range of tissues including immune-privileged sites. These data will provide a high-resolution look at disease kinetics systemically to aid targeted medical countermeasure development and testing.

## Figures and Tables

**Figure 1 viruses-12-00753-f001:**
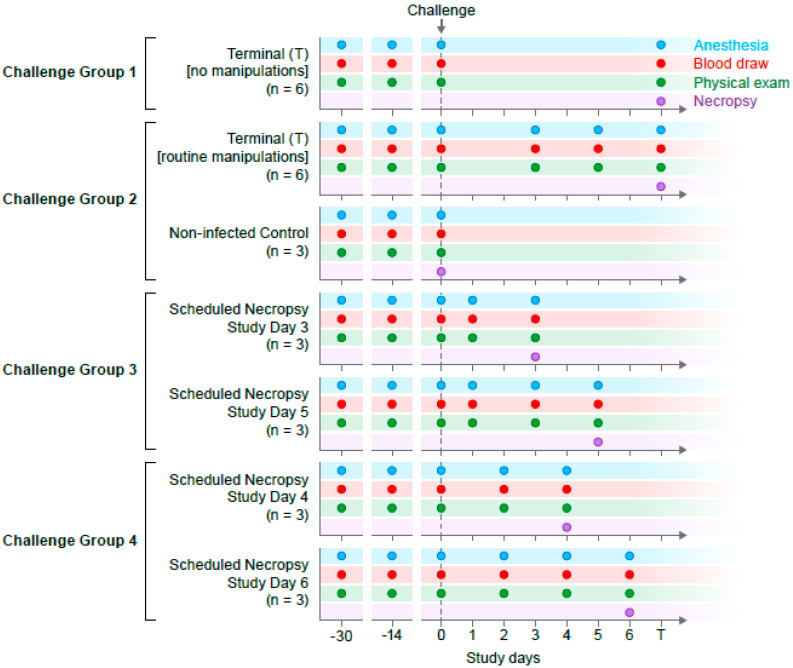
Study design and sample collection timeline. Virus exposure occurred on four dates (challenge groups 1–4) with sample collection based on smaller cohorts. Anesthesia was required for all sample collection activities. Staggered blood draws resulted in a minimum of six individual samples per time point.

**Figure 2 viruses-12-00753-f002:**
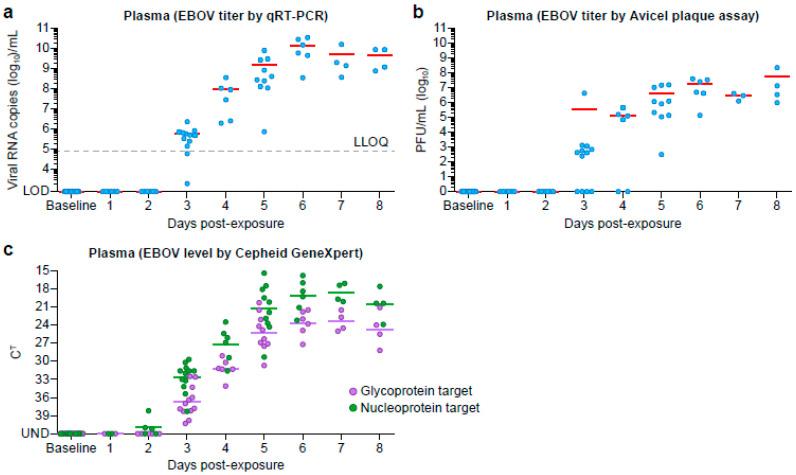
Viral Titer in Blood Products. Viral titers from plasma samples were determined using (**a**) conventional RT-qPCR and (**b**) plaque assay. (**c**) Viral titers from whole blood were determined using the Cepheid GeneXpert.

**Figure 3 viruses-12-00753-f003:**
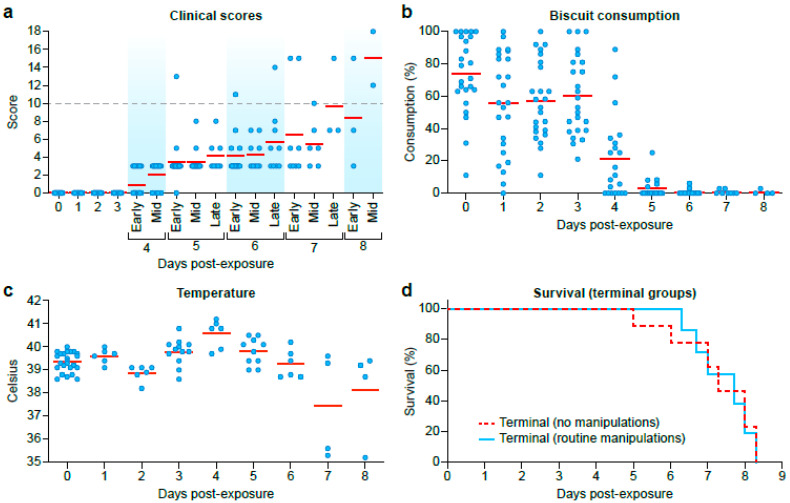
Clinical Observations and Survival for Exposed Animals. (**a**) Clinical scores were collected at least once daily. (**b**) Biscuit consumption was determined once daily. (**c**) Rectal temperatures collected on sample collection time points (**d**) Survival curves for animals allowed to reach predetermined endpoint criteria with EVD that were anesthetized for serial in-life sample collections during the disease course (green) and those that were not (blue).

**Figure 4 viruses-12-00753-f004:**
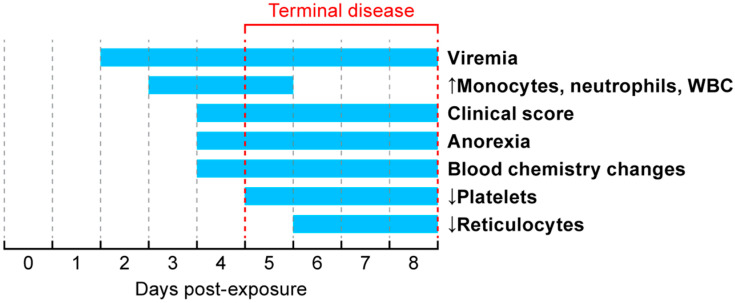
Timeline for major events in Ebola virus disease progression in nonhuman primates. All animals reached predetermined end-point criteria by DPE 8.

**Table 1 viruses-12-00753-t001:** Serum Chemistry Average serum chemistry values and standard deviations are listed with values in red falling outside our internal reference range for that parameter.

Parameter (Units)		Study Day (Number of Animals)										*p* Value ^a^
		BL1 (27)	BL2 (27)	1 (6)	2 (6)	3 (11)	4 (6)	5 (10)	6 (6)	7 (4)	8 (4)	
ALT (U/L)	Avg	36.2	35.5	47.2	44.7	43.2	42.0	189.1	295.2	508.0	370.3	0.260
	SD	9.5	10.8	10.8	12.9	12.3	10.3	183.5	200.4	437.6	228.3	
ALB (g/dL)	Avg	3.5	3.4	3.4	3.4	3.3	3.4	2.6	2.4	1.9	2.2	0.078
	SD	0.3	0.3	0.2	0.2	0.4	0.2	0.4	0.4	0.2	0.1	
ALB:GLOB (ratio)	Avg	0.99	1.03	0.96	0.92	1.01	0.89	0.72	0.62	0.54	0.57	0.092
	SD	0.12	0.12	0.08	0.11	0.22	0.11	0.16	0.14	0.06	0.03	
ALP (U/L)	Avg	164.1	151.4	142.2	147.0	165.6	168.3	299.1	600.2	1074.3	853.8	0.900
	SD	83.0	69.6	39.2	59.3	60.2	49.6	130.8	262.7	484.5	482.5	
AMY (U/L)	Avg	397	385.8	405.7	394.2	336.9	323.3	225.1	382.8	395.5	373.5	0.743
	SD	129.7	138.6	121.7	55.7	78	66.6	72.0	194.7	186.6	197.0	
AST (U/L)	Avg	35.8	34.2	65.3	36.5	43.4	49.0	478.0	844.3	747.3^b^	1221.5	0.972
	SD	6.0	5.8	18.4	5.7	18.4	13.1	462.8	427.3	64.6	506.2	
BUN (mg/dL)	Avg	18.41	19.22	16.8	17.0	16.5	15.8	22.7	29.7	117.8	74.7^b^	0.352
	SD	3.56	4.04	3.6	2.4	3.9	1.5	25.4	9.4	66.6	59.8	
BUN:CRE^c^ (ratio)	Avg	22.3	23.8	30.1	26.5	20.2	15.8	12.3	15.3	9.1	18.2	0.944
	SD	5.8	4.8	5.9	5.0	5.2	2.2	3.9	6.0	9.1	2.2	
CAL (mg/dL)	Avg	9.5	9.4	9.8	9.8	9.2	9.2	8.2	7.7	6.0	8.1	0.258
	SD	0.4	0.3	0.2	0.3	0.4	0.3	0.6	0.5	1.3	0.4	
CRE (mg/dL)	Avg	0.85	0.81	0.57	0.65	0.83	1.02	1.63	2.27	7.28	3.6	0.772
	SD	0.13	0.11	0.09	0.05	0.14	0.13	1.27	1.43	4.68	2.86	
GGT (U/L)	Avg	65.9	61.9	62.5	69.0	64.4	64.7	97.9	159.7	277.3	221.8	0.978
	SD	21.0	20.7	16.9	18.4	12.8	11.4	73.2	56.9	81.3	63.9	
GLOB (g/dL)	Avg	3.60	3.36	3.62	3.75	3.34	3.90	3.78	4.02	3.55	3.88	0.426
	SD	0.36	0.31	0.34	0.39	0.35	0.52	0.69	0.51	0.34	0.27	
GLU (mg/dL)	Avg	67.78	65.78	76.67	72.33	73.82	74.83	69.00	69.67	65.75	45.5	0.275
	SD	11.16	9.64	10.75	4.71	12.81	10.84	14.15	17.13	26.94	21.51	
TBIL (mg/dL)	Avg	0.42	0.42	0.43	0.45	0.42	0.57	0.51	0.85	2.63	1.3	0.823
	SD	0.07	0.06	0.05	0.05	0.04	0.14	0.05	0.34	1.39	0.74	
TP (g/dL)	Avg	7.13	6.78	7.05	7.17	6.63	7.3	6.41	6.45	5.45	6.08	0.830
	SD	0.53	0.44	0.45	0.41	0.35	0.62	0.72	0.56	0.46	0.31	

^a^ The *p*-value comparing terminal chemistry values between groups with (*n* = 6) or without routine manipulations (*n* = 6) during disease course. ^b^ Only three animals had reportable values for this parameter on this day. ^c^ No reference range defined. Abbreviations: “BL”, baseline; “Avg”, average; “SD”, standard deviation; “ALT”, alanine transpeptidase; “ALB”, albumin; “GLOB”, globulin; “ALP”, alkaline phosphatase; “AMY”, amylase; “AST”, aspartate aminotransferase; “BUN”, blood urea nitrogen; “CRE”, creatinine; “CAL”, calcium; “GGT”, gamma-glutamyl transpeptidase; “GLU”, glucose; “TBIL”, total bilirubin; “TP”, total protein.

**Table 2 viruses-12-00753-t002:** Hematology. Average hematology values with standard deviations are listed. Values in red are outside the reference interval for that parameter.

Parameter (Units)		Study Day (Number of Animals)										*p* Value ^a^
		BL1 (27)	BL2 (27)	1 (6)	2 (6)	3 (11)	4 (6)	5 (10)	6 (6)	7 (4)	8 (4)	
Basophils (cells × 10^3^)	Avg	0.01	0.01	0.03	0.02	0.01	0.02	0.01	0.01	0.03	0.15	0.811
	SD	0.01	0.01	0.02	0.02	0	0.01	0.01	0.01	0.04	0.19	
Eosinophils (cells × 10^3^)	Avg	0.08	0.07	0.08	0.01	0.04	0	0.02	0.02	0.01	0.02	0.066
	SD	0.07	0.16	0.07	0.01	0.07	0	0.02	0.01	0.01	0.01	
Hemoglobin (g/dL)	Avg	11.9	11.2	11.8	11.6	11.4	11.5	10.3	11.1	10.8	11.8	0.746
	SD	0.93	1.06	0.3	0.68	0.77	1.03	0.52	1.03	1.77	0.44	
Hematocrit (%)	Avg	37.1	34.9	36.6	35.6	35.1	35.2	31.3	33.6	32.9	35.8	0.487
	SD	2.95	3.34	1.18	2.33	1.81	2.86	1.26	3.62	5.46	1.13	
Lymphocytes (cells × 10^3^)	Avg	2.59	2.56	2.05	3.03	1.7	1.32	1.77	1.57	2.07	2.11	0.741
	SD	1.1	0.75	0.47	1.14	0.59	0.18	1.29	0.43	0.56	1.06	
MCH (pico)	Avg	24.2	24.3	25.2	25.2	24.8	24.7	24.5	23.5	24.7	24.5	0.192
	SD	1.45	1.45	0.99	0.86	1.22	1.18	0.96	1.32	2.00	0.57	
MCHC (g/dL)	Avg	32.2	32.0	32.2	32.6	32.3	32.7	32.8	33.1	33.0	32.9	0.423
	SD	0.8	0.56	0.67	0.41	0.64	0.45	0.72	0.78	0.23	1.12	
MCV (femto)	Avg	75.2	75.8	78.4	75.8	76.8	75.6	74.6	71.0	75.0	74.5	0.784
	SD	4.17	4.29	2.31	3.64	3.35	3.78	3.39	3.43	5.89	1.97	
MPV (femto)	Avg	10.9	10.4	10.7	10.1	10.8	10.6	9.7	10.2	4.9	7.6	0.876
	SD	0.76	0.86	0.75	0.95	0.77	1.00	3.25	0.38	4.91	4.40	
Monocytes (cells × 10^3^)	Avg	0.33	0.39	0.37	0.33	0.78	0.66	0.27	0.15	0.07	0.43	0.669
	SD	0.15	0.14	0.09	0.21	0.25	0.37	0.6	0.26	0.07	0.52	
Neutrophils (cells × 10^3^)	Avg	3.39	4.94	2.68	2.61	8.56	16.49	8.63	6.64	4.05	5.74	0.784
	SD	1.94	3.26	1.17	0.68	2.71	3.5	3.76	4.11	3.36	1.03	
Platelets (cells × 10^3^)	Avg	301.9	315	286.8	314.5	253.4	227.2	129.4	105.2	48.3	52.8	0.414
	SD	58.93	59.21	46.33	61.56	39.58	59.31	28.49	41.76	31.08	4.82	
PDW (femto)	Avg	11.71	11.09	11.33	11.18	11.77	11.5	12.21	13.23	12.6	14	0.327
	SD	1.13	1.36	1.13	1.51	1.3	1.62	0.99	1.72	1.7	2.21	
P-LCR (%)	Avg	32.48	28.14	30.77	26.27	31.76	29.53	31.03	28.15	25.85	27.17	0.095
	SD	6.24	7.15	6.02	8.03	6.11	8.07	3.89	3.37	4.55	2.89	
Plateletcrit (%)	Avg	0.33	0.33	0.3	0.31	0.27	0.24	0.14	0.11	0.07	0.05	0.607
	SD	0.06	0.06	0.05	0.04	0.04	0.05	0.03	0.04	0.04	0	
Reticulocytes (cells × 10^6^)	Avg	0.05	0.05	0.06	0.05	0.04	0.03	0.02	0.01	0.01	0.00	0.071
	SD	0.02	0.02	0.03	0.03	0.01	0.01	0.01	0.00	0.00	0.00	
WBC (cells × 10^3^)	Avg	6.40	8.05	5.22	6.40	11.1	18.48	10.7	8.39	6.22	8.34	0.301
	SD	2.35	3.24	1.03	1.73	2.94	3.69	4.64	4.14	3.57	1.88	

^a^ The *p*-value comparing terminal hematology values between groups with (*n* = 6) or without (*n* = 6) routine manipulations during disease course. Abbreviations: “BL”, baseline; “MCH”, mean corpuscular hemoglobin; “MCHC”, mean corpuscular hemoglobin concentration; “MCV”, mean corpuscular volume; “MPV”, mean platelet volume; “PDW”, platelet distribution width; “P-LCR”, platelet large cell ratio; “WBC”, white blood cells.

**Table 3 viruses-12-00753-t003:** Ebola virus RNA detection in semen. Semen samples collected at baseline or necropsy were tested for Ebola virus RNA using the GeneXpert.

Challenge Group	Cohort	Male Animal	GeneXpert Ct Values (GP, NP)
1	Non-infected Controls	NHP C3	U ^a^
	Terminal Necropsy (No Manipulations)	NHP1	37.3, 35.5
		NHP4	33.9, 32.4
		NHP5	33.9, 29.1
2	Terminal Necropsy (Routine Manipulations)	NHP9	32.1, 28.1
		NHP11	36.4, 33.1
		NHP12	U
3	Scheduled Necropsy (3 Days Post-infection	NHP13	U
4	Scheduled Necropsy (4 Days Post-infection)	NHP20	0, 0 ^b^
		NHP21	0, 0 ^b^
	Scheduled Necropsy (6 Days Post-infection)	NHP23	U

^a^ U (unsuccessful) indicates semen sample was not successfully collected and testing was not done. ^b^ Only a very small residue of semen was collected.

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
