# Peer review of "Kikwit Ebola Virus Disease Progression in the Rhesus Monkey Animal Model"

_viruses, 2020, doi:10.3390/v12070753_

Round 1
Reviewer 1 Report
Importance of the research:
Demonstrating, with a detailed analysis, that anesthesia events for blood draws does not impact the disease, is critically valuable for future studies. Using catheters for blood draws in EBOV studies puts animals through an invasive surgery prior to the challenge, are prone to failures, and can be cumbersome to keep patent. This is an extremely useful finding that can inform study design for future filovirus studies. Overall, the disease kinetics are nicely and thoroughly defined.
Weaknesses of the Paper:
1) No temperature data is shown. In terms of parameters of EBOV disease progression, this is a very important parameter. Do the authors have rectal temperature data that could be used? Could this then be added to Figure 2?
2) No coagulation data is shown. Do the authors have this, even if for only some iterations? If so, this should also be added to the timeline of major events.
3) The important finding/conclusion of anesthesia events not impacting the disease course could be further elaborated on/emphasized. Instead, in the second to last sentence of the manuscript, the authors state that in a MCM efficacy/trigger to treat experiment "daily blood collection would likely only be possible through the use of implanted catheters". I feel that the rationale for this statement needs to be explained more thoroughly in light of conclusion of the research. Do the authors feel that the greater frequency of blood collection would impact the disease course?
4) The wrong figure is shown for Figure 2. (The Figure for Figure 3 is shown twice.)
Minor comments:
5) Line 404 typo: "Ongoing research is focus on"
6) Describe how iterations were conducted, if groups run separately.
7) Figure 3b has a sub-title "Biscuit Consumption (All Animals)". Should this state EBOV exposed animals, since the mock animals were not included on this graph?
8) Some graphs have the x-axis labeled as "Study Day" and some graphs are labeled "Days post-exposure". Prefer "Days post-exposure".
9) Oral swab data as described on lines 205-206 is interesting and it would be nice to see this data as a figure or supplemental figure.
10) Lines 218-219: What percentage of animals had the rash on face and arms at 5-8 DPE? Was the rash on the body/trunk also?
11) Line 244-245: Statement about AST:ALT ratio- does this apply to all animals?
12) Line 263: The "b" in "began" is in a smaller font.
13) Lines 303-304: states "no statistically significant differences" for hematological parameters. I did not see a similar statement for chemistries. Can a similar statement be added?
14) Were gender differences analyzed/ruled out?
15) Sometimes "Challenge Group" (Figure 1) is used and sometimes "Cohort" (Table 3). Consider using one term or defining these terms.
16) Line 386: "Under BSL4 conditions, animals must be anesthetized prior to collection of these clinical hematology and chemistry samples". Catheters can be used (see reference #19) but it is difficult and catheters can fail, complicating the study design. Animals also undergo an invasive surgery which can impact the health of the animals (coagulation parameters can be altered as a result of the surgery). These lines could clarified in a way to drive home one of the important conclusions of the paper. And, the authors could discuss whether an approach avoiding catheters should be considered for pivotal studies.
17) Line 399: "Almost universally lethal"- Could comment on cynos vs rhesus. In the author's research there were no rhesus survivors, but survivors in rhesus have been noted with some frequency (~5-10%) (see reference #19).
Author Response
Reviewer 1
Demonstrating, with a detailed analysis, that anesthesia events for blood draws does not impact the disease, is critically valuable for future studies. Using catheters for blood draws in EBOV studies puts animals through an invasive surgery prior to the challenge, are prone to failures, and can be cumbersome to keep patent. This is an extremely useful finding that can inform study design for future filovirus studies. Overall, the disease kinetics are nicely and thoroughly defined.
Response: Thank you. We appreciate you spending the time reviewing our paper.
Weaknesses of the Paper:
- No temperature data is shown. In terms of parameters of EBOV disease progression, this is a very important parameter. Do the authors have rectal temperature data that could be used? Could this then be added to Figure 2?
Response: Temperature data has been added to Figure 2 as well as into a Supplemental table.
2) No coagulation data is shown. Do the authors have this, even if for only some iterations? If so, this should also be added to the timeline of major events.
Response: We do not have coagulation data to include in this paper.
3) The important finding/conclusion of anesthesia events not impacting the disease course could be further elaborated on/emphasized. Instead, in the second to last sentence of the manuscript, the authors state that in a MCM efficacy/trigger to treat experiment "daily blood collection would likely only be possible through the use of implanted catheters". I feel that the rationale for this statement needs to be explained more thoroughly in light of conclusion of the research. Do the authors feel that the greater frequency of blood collection would impact the disease course?
Response: We’ve added the additional clarification into the paper.
In the present study, short sedation periods used while infected animals were undergoing blood collections did not impact the time-to-death in this study. However, the study design allowed for a 48 hour recovery period between sedation periods and only one blood collection during terminal disease. Daily, or more frequent blood collection, would likely only be possible through the use of implanted catheters due to concerns that frequent sedation could impact animal behavior or feeding and drinking during critical disease timepoints. In addition, as organ function begins to decrease during infection, eliminating the sedatives from the body become more difficult for the animals.
4) The wrong figure is shown for Figure 2. (The Figure for Figure 3 is shown twice.)
Response: We apologize for this mistake, the correct figures have been included.
Minor comments:
5) Line 404 typo: "Ongoing research is focus on"
Response: Changed
6) Describe how iterations were conducted, if groups run separately.
Response: The following statement was added to address this concern. “Challenge groups 1-4 were exposed to virus on separate days due to logistic reasons. Each exposure day used a fresh vial of virus.”
7) Figure 3b has a sub-title "Biscuit Consumption (All Animals)". Should this state EBOV exposed animals, since the mock animals were not included on this graph?
Response: The figure title has been modified to read “Clinical Observations and Survival For Exposed Animals” since we didn’t include non-infected animals in the figure data.
8) Some graphs have the x-axis labeled as "Study Day" and some graphs are labeled "Days post-exposure". Prefer "Days post-exposure".
Reponses: We agree and changed the x axis to “days post-exposure”.
9) Oral swab data as described on lines 205-206 is interesting and it would be nice to see this data as a figure or supplemental figure.
Response: We’ve added the data into a supplemental table.
10) Lines 218-219: What percentage of animals had the rash on face and arms at 5-8 DPE? Was the rash on the body/trunk also?
Response: This was a great question and we added the rash data, collected during cage side observations, into supplemental materials. All animals that reach terminal disease eventually develop a rash on the face (most commonly) and sometime face and arms.
11) Line 244-245: Statement about AST:ALT ratio- does this apply to all animals?
Response: Since the statement only applied to a few animals, we’ve removed the statement so not to confuse the reader.
12) Line 263: The "b" in "began" is in a smaller font.
Response: We have double checked the font size in the word document, but couldn’t identify this issue.
13) Lines 303-304: states "no statistically significant differences" for hematological parameters. I did not see a similar statement for chemistries. Can a similar statement be added?
Response: A similar statement was added for chemistries. Thank you for catching that omission.
14) Were gender differences analyzed/ruled out?
Response: We’ve included the following statement: “In addition, there was no difference in survival time between male or female animals anesthetized for blood collection during the disease course and male and female animals that had no anesthesia (males, p = 0.1628. females, p = 0.1912 by t-test).”
15) Sometimes "Challenge Group" (Figure 1) is used and sometimes "Cohort" (Table 3). Consider using one term or defining these terms.
Response: We added an additional line to the Figure 1 capture defining Challenge Group and Cohort. The terminology was updated in Table 3 as well.
16) Line 386: "Under BSL4 conditions, animals must be anesthetized prior to collection of these clinical hematology and chemistry samples". Catheters can be used (see reference #19) but it is difficult and catheters can fail, complicating the study design. Animals also undergo an invasive surgery which can impact the health of the animals (coagulation parameters can be altered as a result of the surgery). These lines could clarified in a way to drive home one of the important conclusions of the paper. And, the authors could discuss whether an approach avoiding catheters should be considered for pivotal studies.
Response: We’ve added the additional clarification below into the paper.
In the present study, short sedation periods used while infected animals were undergoing blood collections did not impact the time-to-death in this study. However, the study design allowed for a 48 hour recovery period between sedation periods and only one blood collection during terminal disease. Daily, or more frequent blood collection, would likely only be possible through the use of implanted catheters due to concerns that frequent sedation could impact animal behavior or feeding and drinking during critical disease timepoints. In addition, as organ function begins to decrease during infection, eliminating the sedatives from the body become more difficult for the animals.
Although we agree with the reviewers position that catheters are challenging to work with, we opt not to take a position on them in this paper because they may be the only way to collect samples or deliver treatments on a daily schedule.
17) Line 399: "Almost universally lethal"- Could comment on cynos vs rhesus. In the author's research there were no rhesus survivors, but survivors in rhesus have been noted with some frequency (~5-10%) (see reference #19).
Response: The following statement has been added to address cynos vs rhesus. “Of the two most commonly used nonhuman primate in EBOV research, rhesus monkeys and cynomolgus macaques, rhesus monkeys have a longer terminal disease period (5-9 days, average 6.5) compared to cynomolgus macaques (6-7 days) [15-18]. The longer disease course in rhesus monkeys provides a larger period for testing medical countermeasures as well as understanding EBOV pathogenesis.”
As the reviewer pointed out, in our laboratory the FANG EBOV Kikwit stock was 100% lethal. It is difficult to speculate why the virus has not been 100% lethal in other experiments. By using the same virus stock and following our inoculation procedures, we are hopeful that other labs will have the consistent results we do.
Reviewer 2 Report
This study was designed to evaluate a commercially available EBOV stock in Chinese rhesus monkeys. The resulting data is intended to provide a reference for future studies involving medical countermeasures. The authors performed a very sound and large study; however, there are a few items to address as listed below.
INTRO
Line 59 - add "to" after used
Why did the authors choose rhesus monkeys compared to other NHP species?
METHODS
Line 121 - how many samples had failed internal controls? authors should label these separately from true negatives in the results section of the manuscript.
Line 124 - EBOV is known not to plaque very well. could the authors include images of some of their stained plates in the supplementary info?
Line 125 - viruses are typically diluted in serum-free medium to avoid any inhibition of virus entry. have the authors previously compared Kikwit plaque assays with and without FBS?
Could the authors describe the specific criteria required for euthanasia?
RESULTS
Figure 1 - change "none-infected" to "non-infected"
Figures 2 and 3 have the same image; please correct.
Figure 3 - blue and green are labelled incorrectly in the legend.
Line 220 - remove "and"
Line 235 - the interpretation of serum chemistry should be moved to the discussion. it may also be helpful to indicate which major organ each enzyme is relevant to, i.e. AST is a measure of liver function.
Line 349 - was virus isolation attempted on the semen samples?
Table 2 - define BL1 and BL2?
Given that this is a reference data set, it would be valuable to have the individual data for each monkey, i.e. for viremia, appetite, clinical score etc. This way individual trends and differences between each monkey can also be assessed.
DISCUSSION
Line 358 - change "natural history" to "natural disease progression"
Line 361 - changed "exhaustive" to "detailed"
Line 362 - delete "a"
Line 370 - change "viremia detected" to "viremic detection"
Line 370 - change "as early as" to "at"
Line 371 - change "are" to "were"
Line 383 - meaning of the last sentence is not clear, please rewrite.
Line 402 - change "influencing" to "influence"
Line 404 - change "focus" to "focused"
The changes in hematology, clinical signs and clinical chemistry should be compared to previous studies performed in NHPs. Are the results typical of what has previously been found?
Author Response
Reviewer 2
This study was designed to evaluate a commercially available EBOV stock in Chinese rhesus monkeys. The resulting data is intended to provide a reference for future studies involving medical countermeasures. The authors performed a very sound and large study; however, there are a few items to address as listed below.
INTRO
Line 59 - add "to" after used
Response: Added as suggested.
Why did the authors choose rhesus monkeys compared to other NHP species?
Response: The following statement has been added for clarity. “Of the two most commonly used nonhuman primate in EBOV research, rhesus monkeys and cynomolgus macaques, rhesus monkeys have a longer terminal disease period (5-9 days, average 6.5) compared to cynomolgus macaques (6-7 days) [15-18]. The longer disease course in rhesus monkeys provides a larger period for testing medical countermeasures as well as understanding EBOV pathogenesis.”
METHODS
Line 121 - how many samples had failed internal controls? authors should label these separately from true negatives in the results section of the manuscript.
Response: The Cepheid GeneXpert includes two internal controls, a proprietary “Sample Adequacy control” (SAC) designed to determine the adequacy of human whole blood samples, and a “Cepheid Internal Control” (CIC) that determines whether RT-qPCR was successful. Following exposure, only animals NHP20 and NHP24 were considered negative on day 2 due to an invalid SAC but passing CIC, indicating RT-qPCR was successful but that the samples tested were not human blood samples. The methods and results have been reworded to clarify this issue.
Line 124 - EBOV is known not to plaque very well. could the authors include images of some of their stained plates in the supplementary info?
Response: We have not noticed any issues plaquing this EBOV virus stock. Due to a COVID 19 work stand-down, I am unable to generated additional plaques to photograph at this time.
Line 125 - viruses are typically diluted in serum-free medium to avoid any inhibition of virus entry. have the authors previously compared Kikwit plaque assays with and without FBS?
The plaque assay used in this study is modified from Shurtleff et al. (2012) which utilized serum during the dilution of Ebola virus Kikwit. No additional optimization was completed in our laboratory.
Shurtleff AC, Biggins JE, Keeney AE, Zumbrun EE, Bloomfield HA, Kuehne A, Audet JL, Alfson KJ, Griffiths A, Olinger GG, Bavari S; Filovirus AnimalNonclinical Group (FANG) Assay Working Group. Standardization of the filovirusplaque assay for use in preclinical studies. Viruses. 2012 Dec 6;4(12):3511-30.doi: 10.3390/v4123511. Review. PubMed PMID: 23223188; PubMed Central PMCID:PMC3528277.
If the reviewer could share, we’d be interested in a reference for a serum free plaque assay for Ebola virus.
Could the authors describe the specific criteria required for euthanasia?
Response: We have included the euthanasia criteria in Supplemental Table S2.
RESULTS
Figure 1 - change "none-infected" to "non-infected"
Response: Changed
Figures 2 and 3 have the same image; please correct.
Response: We apologize for this mistake, the correct figures have been included.
Figure 3 - blue and green are labelled incorrectly in the legend.
Response: Corrected
Line 220 - remove "and"
Response: Removed
Line 235 - the interpretation of serum chemistry should be moved to the discussion. it may also be helpful to indicate which major organ each enzyme is relevant to, i.e. AST is a measure of liver function.
Response: If the reviewer provided additional information on the interpretation they are concerned about, we’d be happy to review. We have included some statements such as “Serum chemistry results were generally within normal reference interval values (RI) for nearly all analytes evaluated until DPE 4 (Table 1; individual values in Supplemental Tables S3–S17).” To help keep the data in context of what is expected and when the values deviate from the normal range.
We did not explicitly indicate which serum chemistries were directly related to which organ since overall disease can impact these values through multiple pathways. For example, BUN/Cre reflect renal function (glomerular filtration rate), but are significantly dependent on extra-renal factors (blood pressure, urethral patency).
Line 349 - was virus isolation attempted on the semen samples?
Response: Due to small sample volume, we did not attempt virus isolation.
Table 2 - define BL1 and BL2?
Response: Added definition of Baseline.
Given that this is a reference data set, it would be valuable to have the individual data for each monkey, i.e. for viremia, appetite, clinical score etc. This way individual trends and differences between each monkey can also be assessed.
Response: We agree and have added individual data for viremia, biscuit consumption, clinical scores as well as oral virus detection into supplemental materials.
DISCUSSION
Line 358 - change "natural history" to "natural disease progression"
Response: Changed
Line 361 - changed "exhaustive" to "detailed"
Response: Changed
Line 362 - delete "a"
Response: Deleted
Line 370 - change "viremia detected" to "viremic detection"
Response: Changed as suggested.
Line 370 - change "as early as" to "at"
Response: Changed
Line 371 - change "are" to "were"
Response: Changed
Line 383 - meaning of the last sentence is not clear, please rewrite.
Response: The last sentence was modified to read as follows: Ongoing research is focused on how the earliest marker of disease, viremia, quickly establishes a systemic infection connected to establishing infection in a wide range of tissues including immune privileged sites. These data will provide a high-resolution look at disease kinetics systemically to aid targeted medical counter measure development and testing.
Line 402 - change "influencing" to "influence"
Response: Changed
Line 404 - change "focus" to "focused"
Response: Changed
The changes in hematology, clinical signs and clinical chemistry should be compared to previous studies performed in NHPs. Are the results typical of what has previously been found?
Response: Yes, the results are typical of previous EBOV infection studies. We have included the raw data for easy reference which we believe can make this manuscript easier to interpret and reference than some of the original EBOV manuscripts. We have included the following statement in the discussion, “EVD progression in animals on this study was similar to previously reported studies [21-23].”